

# Evaluation of two common source estimation measurement strategies using large-eddy simulation of plume dispersion under neutral atmospheric conditions

Anja Ražnjević[1], Chiel van Heerwaarden[1], and Maarten Krol[1,2]

[1]Meteorology and Air Quality Group, Wageningen University, Wageningen, the Netherlands
[2]Institute for Marine and Atmospheric Research, Utrecht University, Utrecht, the Netherlands

**Correspondence:** Anja Ražnjević (anja.raznjevic@wur.nl)

**Abstract.** This study uses large-eddy simulations (LES) to evaluate two widely-used observational techniques that estimate point source emissions. We evaluate the use of car measurements perpendicular to the wind direction and the commonly used Other Tracer Method 33A (OTM33A). The LES study simulates a plume from a point source released into a stationary, homogeneous and neutral atmospheric surface layer over flat terrain. This choice is motivated by our ambition to validate the

observational methods under controlled conditions where they are expected to perform well since the sources of uncertainties are minimized. Three plumes with different release heights were sampled in a manner that mimics sampling according to car transects and the stationary OTM33A method. Subsequently, source strength estimates are compared to the true source strength used in the simulation. Standard deviations of the estimated source strengths decay proportionally to the inverse of the square root of the number of averaged transects, showing statistical independence of individual samples. The analysis shows that for

the car transect measurements at least 15 repeated measurement series need to be averaged to obtain a source strength within 40% of the true source strength. For the OTM33A analysis, which recommends measurements within 200 m from the source, the estimates of source strengths have similar values close to the source, which is caused by insufficient dispersion of the plume by turbulent mixing close to the source. Additionally, the derived source strength is substantially overestimated with the OTM33A method. This overestimation is driven by the proposed OTM33A dispersion coefficients, which are too large for

this specific case. This suggests that the conditions under which the OTM33A dispersion constants were derived, were likely influenced by motions with length scales beyond the scale of the surface layer. Lastly, our simulations indicate that, due to wind-shear effects, the position of the time-averaged centerline of the plumes may differ from the plume emission height. This mismatch can be an additional source of error if a Gaussian plume model (GPM) is used to interpret the measurement. In case of the car transect measurements, a correct source estimate then requires an adjustment of the source height in the GPM.

## 1 Introduction

Reducing methane emissions can have a more immediate positive influence on the mitigation of climate change than reducing the emissions of carbon dioxide (e.g. Baker et al. (2015); Zickfield et al. (2017); Caulton et al. (2018)). However, methane is emitted by a high variety of activities, which makes the identification and quantification of the sources a complicated endeavour,



and as such the methane budget is uncertain (e.g. Saunois et al. (2016)). In order to address the urgency in constraining the
methane budget, the Methane goes Mobile - Measurements and Modelling (MEMO$^2$) project, in which our study is embedded,
started in 2017. Reducing the uncertainties has two elements. First, methane sources need to be identified, and second, accurate
measurements are needed to quantify the source magnitude. In this paper, we focus on the latter and demonstrate how three-
dimensional large eddy simulations (LES) can help us in estimating the uncertainty in methods that derive the source strength
from field observations, and in setting up appropriate measurement strategies.

Before presenting our study, we provide an overview of the state-of-the art in plume measurement techniques and three-
dimensional simulation of dispersion. Observation of plumes can be performed using a wide range of techniques, such as
satellite remote sensing (e.g. Houweling et al. (2014); Wunch et al. (2019)) and aircraft measurements (Cui et al., 2019),
sensors mounted on towers (Röckmann et al., 2016), unmanned aerial vehicles (UAVs) (Berman et al., 2012; Andersen et al.,
2018; Shah et al., 2019), and tracer release correlation techniques (Mønster et al., 2015; Mitchell et al., 2015). For the detection
and quantification of local sources, techniques using mobile platforms are particularly useful, since they do not require direct
access to the source. In this paper, we will analyze two prominent methods that are widely used: line observations made using
driving cars, and stationary observations using the Other Test Method 33A (U.S. EPA, 2014). In these methods, the Gaussian
plume model is used to translate observations of concentration and wind speed into an expected emission source strength.

The first method, car measurements, has been used for large variety of sources, i.e. leaks from gas and oil production facilities
(e.g. Yacovitch et al. (2015); Atherthon et al. (2017); Baillie et al. (2019)), emissions from landfills (e.g. Hensen & Scharff
(2001)), urban pipeline leaks (Phillips et al., 2013), and agricultural emissions (Hensen et al., 2006). Drawbacks of this method
are its dependency on the available road infrastructure, the necessity to know the exact source location and the assumption
of constant wind speeds that often needs to be made (e.g. Seineld (1986); Atherthon et al. (2017); Caulton et al. (2018)) The
second method, the OTM33A method, combines downwind point measurements of methane concentrations and wind to derive
the emission flux employing the Gaussian plume model. The OTM33A method has been used in estimation of emissions
from oil and gas production facilities (e.g. Brantley et al. (2014); Lan et al. (2015); Foster-Wittig et al. (2015); Robertson et
al. (2017)) and the method has recently been evaluated by Edie et al. (2020). The advantage of this method is its relatively
simple measurement process that relies on wind direction variations that move the plume over the stationary measurement
device positioned directly downwind of the source. By averaging over a sufficient amount of time, a one-dimensional Gaussian
profile of the plume can be recorded. Drawbacks of this method are its inability to account for buoyant plumes, variation in
the emission and measurement heights and ground reflection of the plume. In particular, the latter requirement demands the
observations to be done close to the source on distances of 20 - 200 m (U.S. EPA, 2014; Edie et al., 2020).

Three-dimensional simulation techniques, such as large-eddy simulation (LES) and direct numerical simulation (DNS), can
aid in understanding and quantifying the uncertainties in the two measurement methods. In the past, LES and DNS have
been used to study atmospheric dispersion. LES, which parametrizes the smallest scales of turbulence, has been successful in
simulating dispersion at close and moderate distances from the source (e.g. Boppana et al. (2010, 2012); Matheou et al. (2016);
Ardeshiri et al. (2020)). DNS, which resolves all details of the flow, is now becoming affordable for atmospheric studies (e.g.





Branford et al. (2011); Oskouie et al. (2017)), as computers have sufficient power to simulate atmospheric boundary layers with statistics that are slowly becoming Reynolds number independent.

In this study, we evaluate the car and OTM33A methods using LES. Numerous studies have shown that LES is an established tool for studying plume dispersion (e.g. Dosio & de Arellano (2006); Boppana et al. (2012); Ardeshiri et al. (2020); Cassiani et al. (2020)). Due to the high computational costs involved in LES, we limit this study to a turbulent channel flow. This flow, which is representative for the neutral atmospheric surface layer, is one of the most canonical and well-studied cases of atmospheric turbulence. Moreover, the two measurement methods are expected to perform well under neutral atmospheric

conditions. This study, therefore, provides a baseline test for the two measurements methods. The LES represents perfect field conditions, without interference of confounding factors that may degrade the performance of source estimation methods under actual field conditions.

This paper is structured as follows: In Section 2 we shortly describe the Gaussian plume model, since it is the basis for source estimation in the two studied measurement strategies. We also describe in detail the OTM33A and the car sampling

methods. Furthermore, in Section 3, details of our numerical simulation setup are presented, as well as the implementation of Gaussian-shaped sources, which proved to be necessary for his study. In Section 4 the performance of LES is validated against a wind tunnel experiment, described by Nironi et al. (2015). Furthermore, the similarities of time averaged LES plumes and Gaussian plumes are discussed, followed by an analysis of the impact of plume averaging on source strength estimations. Finally, Section 5 provides an overview and discussion of our findings.

## 75  2  Measurement methods

Here, we discuss the two measurement methods: measuring from driving cars and the OTM33A method. Before discussing both in detail, we provide a brief overview of the Gaussian plume model as this is the essential model for both methods to convert concentration measurements into a source strength.

### 2.1  The Gaussian plume model

The simplest approach to describe plume dispersion is the Gaussian plume model, which represents the stationary solution to the advection-diffusion equation (e.g. Seineld (1986)). The solution to the equation with a reflective ground component is given by Eq. 1 (e.g. Csanady (1973)):

$$C(x,y,z) = \frac{Q}{2\pi\sigma_y\sigma_z\overline{u}} \exp\left(-\frac{(y-y_s)^2}{2\sigma_y^2}\right)\left[\exp\left(-\frac{(z-z_s)^2}{2\sigma_z^2}\right) + \exp\left(-\frac{(z+z_s)^2}{2\sigma_z^2}\right)\right], \tag{1}$$

where $C$ is the scalar concentration at the position (x,y,z), and $\sigma_y$ and $\sigma_z$ are the plume dispersion parameters which depend

on the distance from the source and the atmospheric stability. These parameters are calculated following one of many proposed parametrizations, most of which follow the Pasquill-Gifford's stability class scheme (e.g. Seineld (1986); Briggs (1973); Korsakissok et al. (2009)). $Q$ is the source strength positioned at ($x_s = 0$, $y_s$, $z_s$) and $\overline{u}$ is the mean wind along the x-axis. The model





has been studied in detail, and advanced versions are currently in use as a fast-response approach to scalar dispersion modeling (e.g. Cimorelli et al. (2005); Korsakissok et al. (2009)). One of the main assumptions of this model is the plume stationarity,

which deviates greatly from the measured instantaneous plumes, and the model should be interpreted as an average of an infinite number of instantaneous plumes (e.g. Seineld (1986)). Studies suggest that the sufficient averaging time, depending on the distance from the source and the stability, ranges between 2 and 60 minutes (Fritz et al., 2005). For the neutral stability class D, which corresponds with our study, the averaging time ranges from 2 to 30 minutes at distances of 100 to 1000 m.

One set of dispersion coefficients $\sigma_y$ and $\sigma_z$ that is widely used (e.g. Korsakissok et al. (2009) ) is the Briggs parametrization.

This parameterization is appropriate for urban and for rural sites and is given in the form (Griffiths, 1994):

$$\sigma_i = \alpha x (1 + \beta x)^\gamma, \tag{2}$$

where i = (y, z) and $\alpha$, $\beta$ and $\gamma$ are coefficients that depend on the dispersion direction, stability class, and orography of the site where measurements are taken and $x$ [m] is the downwind distance from the source. For rural sites and neutral stability, coefficients have values of $\alpha$ = [0.08, 0.06], $\beta$ = [0.0001, 0.0015] for the y and z directions respectively, and $\gamma$ = - 0.5 for both

directions.

## 2.2    OTM33A Measurement Method

The OTM33A method was developed by the US Environment Protection Agency (U.S. EPA, 2014). The method consists of two parts: detection of plumes and quantification of emissions. The detection is performed by driving downwind from the likely source, perpendicular to the mean wind direction, with the goal to detect the plume centerline. After the plume centerline

is detected, the car is parked directly downwind from the source, at distance x $\in$ [20, 200] m. The inlet of the measurement device is oriented directly towards the mean wind direction in order to minimize the impact of turbulent eddies generated by the measurement equipment. Subsequently, the methane concentrations, wind speed, wind direction, and temperature are measured continuously for 20 min at the assumed height of the release. Emissions are quantified following the Gaussian plume equation, with the assumptions (i) the measurement inlet is positioned at the height of the release (ii) the measurement are

taken directly downwind from the source and (iii) the reflection from the ground is negligible at [20-200] m from the source. Therefore, $Q_{estim}$ [kg s$^{-1}$] can be estimated from:

$$Q_{estim} = 2\pi \sigma_y \sigma_z c_{max} \overline{u}, \tag{3}$$

where $\sigma_y$ and $\sigma_z$ [m] are dispersion coefficients that are provided in look-up tables. These coefficients depend on the distance from the source and the atmospheric stability. To calculate $c_{max}$, methane concentrations are binned in wind-direction bins of

10°, and the average methane concentration in every bin is calculated. $c_{max}$ [kg m$^{-3}$] is taken from the bin with the highest averaged concentration. $\overline{u}$ denotes the average wind speed [m s$^{-1}$] during the measuring period. Note that equation 3 does not have a term that accounts for plume reflection at the surface, buoyancy of the plume, and a possible difference between the source height and the measurement height. Equation 3 assumes no background concentrations.





The OTM33A method is used for the quantification of small (point-like) sources. Since the distances over which this method
is employed are not sufficient for the plume to fully disperse, the plume is still narrow, patchy, and meandering in behavior
(Gifford, 1959). Moreover, the method assumes that the terrain over which the plume is dispersing is flat without any obstacles
that can distort the shape of the plume.

## 2.3   Estimating source strength from car measurements using an Inverse Gaussian method (IGM)

The car measurement method consists of measurements perpendicular to the mean wind direction, downwind from the source
(e.g. Yacovitch et al. (2015)). This method, as opposed to OTM33A, provides the one-dimensional spatial extent of the plume
by moving the instrument instead of relying on the wind direction changes to move the plume over the instrument. Usually,
the meteorological conditions are measured simultaneously by either instruments placed on site (Caulton et al., 2018) or
instruments placed on the car (Atherthon et al., 2017).

The method for estimating the source strength from measured plume transects is based on the ratio of modeled and measured
concentrations. If the mean wind is along the x-axis and drive-by's are in the cross-plume y-direction, then the source strength
can be calculated by summing the modeled and measured concentrations $C_{meas}$ along the y-axis and by scaling the source
strength (Caulton et al., 2018)

$$Q_{estim} = \frac{\sum\limits_{y} C_{meas}}{\sum\limits_{y} C_{Gauss}} \times Q_r \qquad (4)$$

where $C_{Gauss}$ are the modeled concentrations. To calculate $C_{Gauss}$, Eq. 1 is used with the referent emission rate $Q_r$,
measured mean wind $\overline{u}$, and dispersion coefficients chosen for the encountered conditions in the field. The estimates rely on
line integral in the y direction. Therefore, the technique is not sensitive to possible misrepresentation of lateral dispersion in
the modelled plume but assumes that the vertical dispersion is described correctly. Equation 1 takes into account the reflection
from the ground and assumes that the exact location and height of the source are known. In this procedure, no background
concentrations are assumed.

## 3   Case set-up and numerical simulation

Numerical simulations have been performed using MicroHH (www.microhh.org, van Heerwaarden et al. (2017)).

We study a stationary, homogeneous, turbulent channel flow in which a non-reactive scalar is being released from multiple
point sources. The model set-up follows the experimental study by Nironi et al. (2015).

Our simulation uses a second-order-accurate finite volume scheme to solve dynamics in the system. For the advection, sixth
order interpolations are applied and for the advection of scalars a flux limiter is applied to ensure monotonicity. Time is ad-
vanced with a third order Runge-Kutta time-integration scheme. We use periodic boundary conditions for the three wind com-
ponents on the lateral boundaries of the domain. The second-order Smagorinsky model is used for the subgrid parametrization





of the velocity components. The upper boundary condition is free-slip and the tangential components of velocity are assumed zero ($\frac{\partial u}{\partial z} = \frac{\partial v}{\partial z} = 0$). There is no penetration through the upper boundary ($w = 0$). The lower boundary has no-slip ($u = v = 0$)

boundary conditions and no penetration through the lower boundary. For the scalar, in-flow and out-flow conditions were set on all the lateral boundaries to prevent it from re-entering the domain. Dirichlet boundary conditions are set for in-flow on the left and upper boundary and Neumann conditions for the out-flow on the right and lower boundary.

### 3.1  Implementation of sources

The MicroHH code has been extended to support placement of point and line sources of scalars at arbitrary positions in the

domain. In order to avoid numerical artefacts, which would arise from injecting tracer mass into the simulation at a single grid cell, the implementation of a point source is achieved in the form of a 3-D Gaussian function that spans over $[-4\sigma_i, 4\sigma_i]$, where $\sigma_i$ is the standard deviation in the respective coordinate direction (i = x, y, z), around the source location ($x_0$, $y_0$, $z_0$). The value of $\sigma_i$ is chosen by the user, dependent on the required size of the source. Consequently, the source $S$ that is added to the grid has the shape

$$S(x,y,z) = Q \, s \, \exp\left( -\frac{(x-x_0)^2}{\sigma_x^2} - \frac{(y-y_0)^2}{\sigma_y^2} - \frac{(z-z_0)^2}{\sigma_z^2} \right). \tag{5}$$

Here, $Q$ [kg s$^{-1}$] is the total source strength that is released in the simulation, distributed over the 3-D Gaussian function $S(x,y,z)$. The source $S$ integrates into $Q$ by using a normalization constant:

$$s = \frac{1}{\sqrt{\pi^3}\sigma_x\sigma_y\sigma_z\mathrm{erf}(4)^3}. \tag{6}$$

### 3.2  Numerical experiment

As previously mentioned, the domain was set up to mimic the experimental study of Nironi et al. (2015), with a domain size of $6144 \times 1536 \times 1000$ m, and sources were placed at $306 \times 770 \times [0, 60, 190]$ m. The friction velocity had the value $u_\tau = 0.16$ m s$^{-1}$, the eddy viscosity was $\nu = 0.011$ m$^2$ s$^{-1}$ and the wind speed at the top of the domain was $u = 5$ m s$^{-1}$. The domain was discretized on a $1536 \times 384 \times 360$ grid, with uniform spacing in the horizontal direction ($\Delta x = \Delta y = 4$ m), and a stretched grid in the vertical with $\Delta z \approx 1$ m close to the surface and $\Delta z \approx 6$ m at the top. The sources were added into the simulation

as a 3D (elevated sources) or 2D (ground source) Gaussians (section 3.1) with $\sigma_{source} = 4$ m, equivalent to one grid box size. Note that the source at 0 m was not part of the Nironi et al. (2015) experiment. Nevertheless, we add this experiment because ground sources are often encountered when measuring in the field. The source strength for all three sources is set to $Q_{source}$ $= 1 \cdot 10^{-3}$ kg s$^{-1}$. The simulation was first run for 25200 s to achieve statistical stability of the flow, after which the three sources were released into the flow and run for an additional 3600 s. The concentrations were recorded every 1 s on multiple

downwind distances over various 2D domain transects (x − y, x − z or y − z).



### 3.3 Plume sampling

#### 3.3.1 Simulating plume meandering

Figure 1a shows that, when our simulated plume is sampled according to the OTM33A protocol, we capture very narrow plume with wind direction that spans over [-10°, 10°] angle. We hypothesize that this is caused by the absence of large-scale meandering in our simulation. The external forcing of our flow is determined by a large-scale pressure gradient force directed constantly in the x direction of the domain. As a result, our LES only contains meandering motions that are driven by turbulent motions in the domain itself and not by larger scale flow fluctuations. Close to the emission source, the OTM33A sampling protocol always samples the plume. Consequently, the sampled concentration variations visible in Fig. 1a are mostly caused by the applied shape of the emission source (see section 3.1). Small-scale turbulent motions did not have time to mix the plume, which is consequently still retaining the shape of the source. In order to impose the lacking large-scale meandering, we mimicked meandering of the plume by moving the measurement point through the plume, perpendicular to the mean wind.

The sampling was performed on an angle of $\theta \in$ [-15°, 15°] around the plume centerline in the y direction. This angle was chosen in order not to sample outside of the plume. Onto the sampling angle, the instantaneous wind direction measured at each sampled point was added. We move the location at which we record the sample back and forth between the plume edges, with a denser sampling close to the centerline, as shown in Figure 1c (see Appendix A). Note that we impose larger values of $\Delta$y when we sample further downwind from the source. Figure 1 b shows the resulting OTM33A concentrations after we sampled the plume with the additional meandering.

We have sampled the plume at four downwind distances that fall into the proposed range (x = [20, 200] m, Edie et al. (2020)). The samples were taken at x = [48, 108, 152, 200] m from the source. All three plumes were sampled at the height of their release (i.e. [0, 60, 190] m) with the frequency of 1 Hz for a duration of 30 min. Each plume was sampled 20 times with a time delay of 60 s between each sample to achieve reliable statistics.

#### 3.3.2 Simulating car sampling

The sampling of the plume mimicking the car movements was performed in a similar manner as OTM33A measurements. The concentration measurements were taken perpendicular to the mean wind over the whole width of the domain (1536 m). The measurements were taken at the height of the release for each of the three emission heights, and at eight downwind distances from the source, x = [108, 200, 312, 624, 1248, 2500, 3748, 5000] m.

Firstly, we have recorded instantaneous plume transects over the y direction, i.e. mimicking an infinitely fast car. These instantaneous samples are used as a benchmark for plume measurements taken with realistic car speeds. We have taken 70 sets of measurements, each consisting of 100 plume transects to gather statistics. The time delay between each set of measurements was taken as 10 s. The same time delay was applied between each consecutive transect in an individual set. Secondly, to study the possible influence of driving speed on the source strength estimations, we have sampled the plume with two different car speeds, V = [4, 12] m s$^{-1}$, with a sampling frequency of 1 Hz. The 1 Hz frequency represents the highest temporal resolution



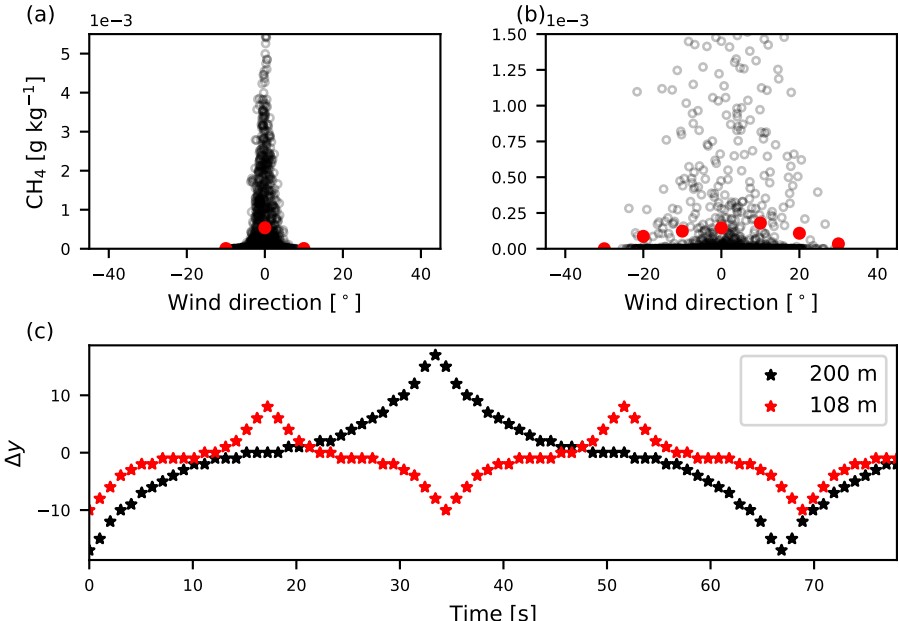

**Figure 1.** (a) Methane concentration plotted against the wind direction according to OTM33A protocol for the case when no meandering is imposed. (b) Methane concentrations against wind direction with imposed meandering. Red circles indicate bin averages. Bins of $10°$ are used. (c) An example of the sampling pattern used to impose meandering for two distances from the source.

available from our simulation. As with the instantaneous plume transects, we recorded 70 sets of 100 plumes, with a time delay of 10 s between sets and individual plumes respectively.

To study the influence of atmospheric variability on the source strength estimation when using the Gaussian plume model, we averaged plumes in each of the 70 sets for each sampling strategy. The averaging was performed such that the resulting plume $\overline{C_j}^t$ (j $\in$ [1, 384] is the position on the y-axis) is an average of $t$ ($t \in$ [1, 100]) previous plumes. In this way, we transformed each set of turbulent plumes into a set of averaged plumes. The first element is a single, non-averaged plume, and the last plume is an average of 100 plumes.

**3.4 Statistical properties of the plumes**

To further our understanding of the processes that govern plume dispersion close- and far-field from the source, plume dispersion can be subdivided into two processes. The first process (relative dispersion) is mixing by the turbulent eddies with a size smaller than or comparable to the size of the plume. The second process (meandering) is the displacement of the plume center of mass by the turbulent eddies that are larger than the size of the plume (e.g. Dinger et al. (2018)).

To separate these two processes, first the center of mass of the instantaneous plume, $z_m$, on its y-z transect is defined as:

$$z_m(x,t) = \frac{\int c(x,y,z,t)\, z\, dz\, dy}{\int c(x,y,z,t)\, dz\, dy}. \tag{7}$$





An ensemble average over many such instantaneous plumes will be equal to the center of mass of the time averaged plume $\overline{z_m}$. Next, different metrics that measure plume displacement from its center of mass are defined. First, the absolute fluctuation $z'$ is the displacement of an in-plume particle from the mean center of mass $\overline{z_m}$. Second, the relative fluctuation $z'_r$ is the

displacement of a in-plume particle from the instantaneous plume center of mass $z_m$. Third, $z'_m$ is the displacement of the instantaneous plume centerline from the mean plume center of mass. These three metrics relate to each other as:

$$z' = z - \overline{z_m}, \quad z'_r = z - z_m, \quad z'_m = z_m - \overline{z_m}. \tag{8}$$

Now the vertical plume widths, stemming from the two dispersion processes $\sigma_{z,mean}$ (meandering) and $\sigma_{z,mix}$ (mixing) are defined as:

$$\sigma^2_{z,meand}(x,t) = \frac{\int c(x,y,z,t)\, z'^2_m \, dy\, dz}{\int c(x,y,z,t)\, dy\, dz}, \quad \sigma^2_{z,mix}(x,t) = \frac{\int c(x,y,z,t)\, z'^2_r \, dy\, dz}{\int c(x,y,z,t)\, dy\, dz}. \tag{9}$$

A similar expression applies to the total plume spread $\sigma_{z,tot}$ around the mean center of mass $z_m$:

$$\sigma^2_{z,tot}(x,t) = \frac{\int c(x,y,z,t)\, z'^2 \, dy\, dz}{\int c(x,y,z,t)\, dy\, dz}, \tag{10}$$

where

$$\sigma^2_{z,tot} = \sigma^2_{z,meand} + \sigma^2_{z,mix}. \tag{11}$$

Similar expressions apply to dispersion in y direction.

## 4  Results

### 4.1  Velocity and mean plume statistics

As a first step, the velocity statistics from the LES are validated against wind tunnel measurements presented in Nironi et al. (2015). The statistics are obtained as time (60 samples over an hour of simulation) and horizontal (over the whole domain)

averages. Figure 2 a shows discrepancies between the non-dimensional wind speeds in the experiment and the LES. The wind speed at the top of the boundary layer in the LES is 4.9 m s$^{-1}$, which is very close to the value of 5 m s$^{-1}$ presented in Nironi et al. (2015). However, the friction velocities, $u_*$, have values of 0.163 m s$^{-1}$ and 0.185 m s$^{-1}$ in the LES and the experiment respectively. As a result, the mean wind speeds differ when normalization with $u_*$ is used. Another possible reason of the discrepancies is the so-called overshoot of the mean wind in LES, which has has been addressed previously (e.g. Brasseur &

Wei (2010); Ardeshiri et al. (2020)). Overshoot in LES has been found to depend on the subgrid-scale (SGS) model, grid aspect ratio, grid resolution and the wall model. Despite the discrepancy in the mean wind, very good agreement is found between the wind speed variances (Fig. 2 b) and covariances (Fig. 2 c). Very good agreement is found for the triplet correlations as well (Fig. 2 d).





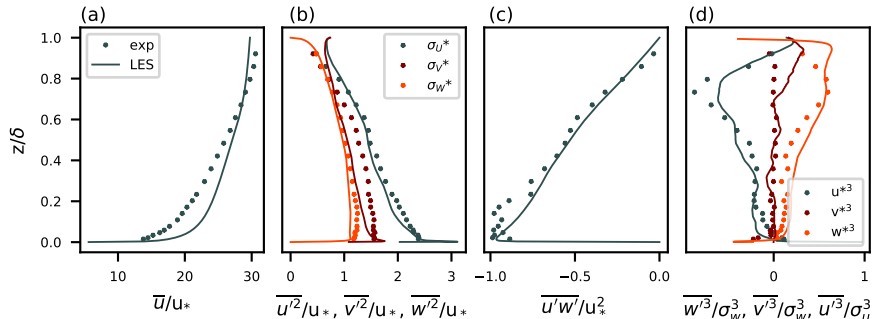

**Figure 2.** Vertical profiles of non-dimensional velocity statistics and comparison with the Nironi et al. (2015) data. (a) Mean longitudinal wind speed, purple line indicates the law of the wall with $u_*$ from Nironi et al. (2015). (b) Variances of three wind components, (c) Reynolds stress and (d) triplet correlations.

Following the good agreement of the higher order velocity statistics, we expect that the mixing of the plume in the cross-wind directions is well represented in the LES. The longitudinal mean wind affects the advection of the plume, i.e. the time the plume spent in the atmosphere being mixed by the turbulent eddies. Consequently, the statistics of the Nironi et al. (2015) plumes and the LES plumes cannot be compared at the same downwind distances. They can, however, be compared at the same effective distances from the source $x_*$, defined as the downwind distance $x$ scaled with one eddy overturn distance $X$:

$$x_* = \frac{x}{X} = \frac{x}{\overline{u}T} = \frac{u_* x}{\overline{u}\delta},$$

(12)

where $T$ is the characteristic eddy overturn time, $\overline{u}$ is the mean wind speed, $u_*$ is the roughness speed and $\delta$ is the boundary layer height. The downwind distance $x_{LES}$ at which the LES plume has spent an equal amount of "mixing-time" compare to the Nironi et al. (2015) plume ($x_N$) is:

$$x_{LES} = \frac{u_{*,N}}{u_{*,LES}} \frac{\overline{u_{LES}}}{\overline{u_N}} \frac{\delta_{LES}}{\delta_N} x_N.$$

(13)

Figure 3 shows a comparison of the first four statistical moments of the mean plume concentrations at the distance $x_N = 2.5$ $\delta$. The moments are calculated over the horizontal and vertical plume transects at the height of the release and the $y$-position of the source respectively. The comparison is shown for the plumes released at $0.06\ \delta$ and $0.19\ \delta$. The moments have been normalized (denoted by superscript $*$) with the plume emission rate $Q$ [g s$^{-1}$], free-stream velocity $u_\infty$ [m s$^{-1}$] and the height of the boundary layer $\delta$, e.g. $C^* = C \frac{u_\infty \delta^2}{Q}$ is the normalized mean plume concentration. The mean plume profiles (Fig. 3 first row) show very good agreement over both transects: peaks of concentration and the plume width are well captured in the LES. The variances for the sources at $0.19\delta$ (Fig. 3 second row) also show very good agreement. For the plume emitted closer to the ground, the variances agree well with the experiment at the edges of the plume, but are higher in the LES in the plume centerline. The same can be observed for the other two moments shown here (Fig. 3 bottom two rows). Note here that, for





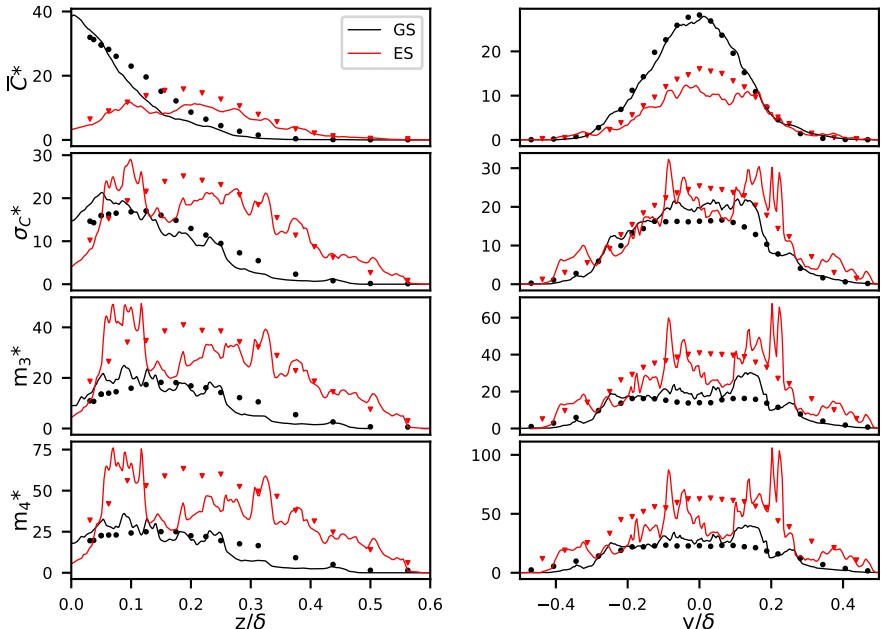

**Figure 3.** Vertical (left column) and horizontal (right column) profiles of the first four statistical moments of plume concentration. Lines denote LES results and symbols are corresponding results from Nironi et al. (2015). GS (black) denotes the source emitted at $0.06\delta$ and ES (red) the source at $0.19\delta$. The transects are taken at $x_N = 2.5\delta$ for the experiments. The corresponding values for $x_{LES}$ are 4.5 and $4.2\delta$ for GS and ES, respectively.

higher moments, LES curves do not show the same smoothness visible in the experiment despite the 600 samples used to calculate the average.

To give the reader an intuitive understanding of the spatial characteristics of the turbulence, Figure 4 shows instantaneous x-z cross-sections of the three simulated plumes at the y position of the source ($y_s$). The lowest plume stays relatively close to the surface and slowly mixes up with the increasing distance from the source. The middle plume stays compact around the emission height for a relatively short time before it is transported towards the surface. In contrast, the highest plume stays elevated for considerable distance from its source ($\approx 3000$ m) before it gets transported to the surface. While elevated, the

highest plume exhibits highly meandering behavior: the spread of the plume around its instantaneous center of mass is narrow, and is transported and broken up by larger eddies.

To illustrate these meandering motions, Figure 5 shows 100 instantaneous y-transects taken at emission height for each plume, separated by 24 s and at 1248 m from the source. Clearly, enhanced variability is found for the highest plume. Large eddies do not cause mixing close to the surface, and dispersion at this level is predominantly caused by diffusive processes.

Furthermore, the lowest plume exhibits higher mean concentrations, which can be attributed to the lower mean wind speed close to the ground (Fig. 2 a).





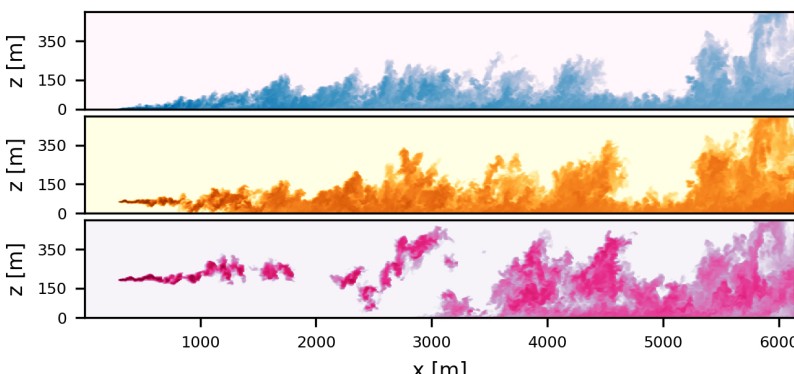

**Figure 4.** Snapshot of x-z transects of the three plumes taken through the plume centerline. Blue, orange, and pink correspond to emission heights of 0, 60, and 190 m, respectively.

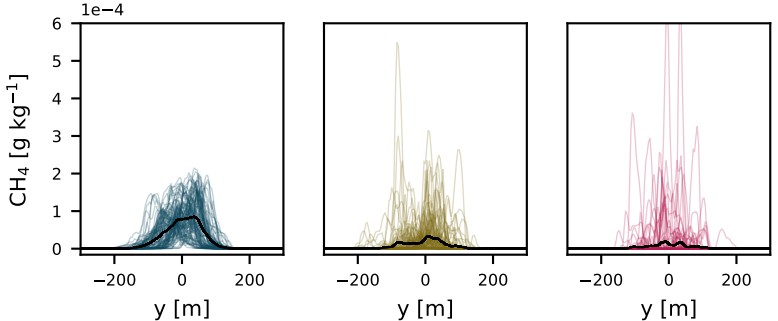

**Figure 5.** Example of 100 instantaneous transects from 3 plumes, taken at the emission height of the respective plumes, at a distance of 1248 m from the source. The mean of the 100 plumes is shown in black. Blue, brown, and pink correspond to emission heights of 0, 60, and 190 m, respectively.

### 4.2 Structure of the time averaged LES plume

The Gaussian plume model is a solution to the stationary advection-diffusion equation (Seineld, 1986), and can be interpreted as an average of an infinite number of plume realizations. Therefore, by time averaging the LES plume over a large number of time steps, a Gaussian plume shape is expected. Figure 6 shows the time-averaged LES plumes in the x-z plane at the y position of the releases ($y_s$ = 0 m). Figure 6 also shows the edges $\sigma_{ztot,LES}$ of the plumes and the plume centerline $\overline{z_m}$ (see section 3.4). For comparison, the edge of a Gaussian plume $\sigma_{z,Briggs}$ for stability class D defined by Equation 2 are given.

Firstly, it can be observed that mean plume centerlines behave differently depending on the release height. For the highest release height (Fig. 6 (bottom)) the mean plume centerline stays at the emission height irrespective of the downwind distance from the source. Conversely, the plume centerline is lifted from the surface for the source at 0 m (Fig. 6 (top)) and at 60 m (Fig. 6 (middle)). This is a consequence of the vertical velocity field that is positively skewed at the lower heights (not shown). As





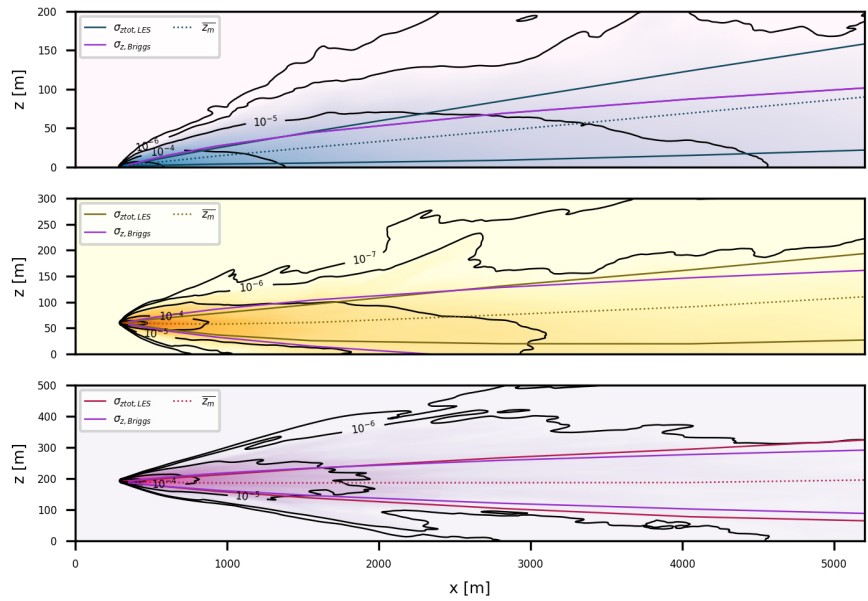

**Figure 6.** Time averaged x-z cross-section of the LES plume at (top) 0 m, (middle) 60 m and (bottom) 190 m height. The concentration fields are averaged over half an hour. Also plotted are the plume edges of the Gaussian plume (assuming Briggs diffusion coefficients) and the LES plumes. Centerlines $\overline{z_m}$ are plotted as dashed lines.

a result, there are large areas of slowly sinking motions with occasionally strong upward ejections lifting the mean centerline position. Secondly, the lowest plume shows clear discrepancies in the lines that outline the plume edges in the Gaussian plume model and the LES. The Gaussian plume model only accounts for the effects of vertical mixing through the vertical dispersion

coefficient, $\sigma_z$. Consequently, the plume centerline remains always at the emission height. Lastly, for the highest emission height, the width of the Gaussian plume and the highest LES plume only start to diverge far from the source. For the lower two emission heights, the differences between the plume widths are larger. This is better illustrated in Fig. 7 b. Here, $\sigma_z$ values from Briggs and the 190 m release height are similar to approximately 1000 m downwind from the source before they start to diverge. The slower vertical dispersion of the lower two plumes is also clearly visible. LES therefore indicates that vertical

dispersion coefficients should be height dependent to capture changes in the wind regime with height. In contrast, horizontal dispersion coefficients (Fig. 7 a) show very little variation with changing release height, but are much smaller than the Briggs Gaussian plume coefficients. The small dispersion in the y direction can be attributed to the lack of the large scale forcing in our simulation or the absence of eddies larger than the domain size (1536 m). We dictated the meandering part of dispersion in the horizontal direction (see Section 3.3.1).

To mimic the Gaussian plume growth in both directions with Eq. 2, Table 1 gives the optimized coefficients that lead to a match with the LES plumes. These coefficients will be used in Section 4.3.1 to compare to the dispersion coefficients from the look-up tables used in the OTM33a method.





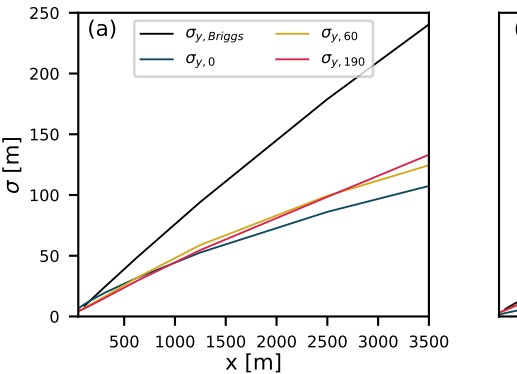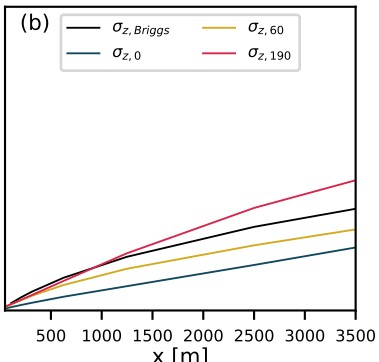

**Figure 7.** (a) Horizontal and (b) vertical plume width as a function of downwind distance from the source. Plume widths are shown for all three release heights as well as plume widths calculated using Briggs (Eq. 2).

**Table 1.** Coefficients of horizontal and vertical plume dispersion (Eq. 2) fitted for LES plumes with different source height. $\gamma = -0.5$ remains unchanged from eq 2.

| Source height [m] | | 0 | 60 | 190 |
|---|---|---|---|---|
| $\sigma_y$ | $\alpha$ | 0.07 | 0.062 | 0.048 |
| | $\beta$ | 0.001 | 0.001 | $2 \times 10^{-4}$ |
| $\sigma_z$ | $\alpha$ | 0.017 | 0.043 | 0.049 |
| | $\beta$ | $9 \times 10^{-5}$ | 0.001 | $5 \times 10^{-4}$ |

Now we move on to use the LES results to evaluate two techniques to infer the source strength from downwind concentration measurements: OTM33A, and the Inverse Gaussian Model using drive-by's.

### 4.3 OTM33A

In order to obtain Gaussian profiles of mean concentrations (see Fig. 1), we followed the sampling procedure described in Section 3.3.1 to mimic plume meandering. Source strength estimates using the OTM33A method at 4 different distances from the source (x = [48, 108, 152, 200] m) are shown in Fig. 8. The tracer concentrations were recorded over 20 min (1200 data points). To obtain measurement ensembles, the sampling was repeated 20 times, as described in Section 3.3.1.

The most striking result is that for nearly all emission heights the OTM33A method overestimates the source strength. Uncertainties generally increase slightly with increasing distance from the source. Only for the 190 m source the estimated source strength at the downwind distance of 48 m is close to the true source strength. We found that this effect is caused by the way we introduce the source in the atmosphere. Instead of emitting the tracer as a point source, we use a Gaussian distribution to pre-disperse the source (see sections 3.1 and 3.2). In combination with the implemented meandering (section





3.3.1), sampling close to the source coincidentally leads to plume dispersion similar to the dispersion of the OTM33A imposed Gaussian plume.

Further away from the source and with lower emission heights, a general overestimation is found. The estimated source strengths using the OTM33A method depends linearly on the dispersion parameters (Eq. 3). Consequently, too large dispersion parameters automatically lead to overestimated source strengths. The dispersion parameters used here were taken from the recommended look-up table (U.S. EPA, 2014). These values are based on the Pasquill-Gifford dispersion curves, just like the Briggs coefficients. As we have shown above (Fig. 7 a), these values are too large in the y direction compared to our LES dispersion calculation, explaining the overestimates. We have also shown that dispersion parameters in the z direction depend on the height of the source. While the differences in the z direction are not so pronounced, especially for the highest source (Fig. 7 b), they also contribute to the error in estimates. At the closest distance from the source, the estimates for the two lower sources have larger errors compared to the highest source. There are several causes for this. Firstly, as previously discussed, the vertical dispersion coefficient for the highest plume has a better agreement with the Briggs dispersion coefficient at distances close to the source. Secondly, according to the OTM33a protocol, the concentrations should be recorded at the plume centerline, which is assumed to be at emission height. However, we found that for different emission heights the instantaneous plumes centerline positions behave differently. Figure 9 shows the pdfs of instantaneous plume centerline positions relative to the y and z position of the source ($y_s$, $z_s$). In the z-direction, the pdfs have longer tails for values above the emission height (positive skewness). In contrast, the lowest plume lifts off the ground with distance from the source. With downwind distance, the displacements from the emission height grow as do the errors in the source strength estimates. In the y direction, the highest and the lowest plumes have slightly positively skewed pdfs relative to the emission point. The skewness in the middle plume is even more pronounced. This, in combination with the plumes still being very narrow, results in plumes being sampled at their edges, which leads to high estimation errors. The error also depends on the distance from the source as the height of the plume median is not constant. Further downwind this effect is less pronounced since the plumes get wider and sampling slightly out of the plumes median position still characterises the plume well.

Next, we study the influence of the averaging time on the source estimation. In Fig. 10 we show source estimates for six different averaging times at four sampling distances. The estimates for all emission heights show similar behaviour. Averaging for 20 minutes leads to smaller estimation errors for all three sources. Convergence of the error becomes slower with increasing distance from the source for the two higher sources. The lowest source had a small estimation error even for short averaging times on most downwind distances. From Fig. 5 it is visible that the lowest plume shows little variability even when measured much further from source (1248 m on Fig. 5) than the OTM33a method suggests.

Lower variability in estimations closer to the source can be attributed to short dispersion time. As previously mentioned, the plumes dispersion is a combination of relative dispersion around the instantaneous center of mass and the meandering motions. On very short distances from the source the plume will retain the initial source shape until it gets sufficiently mixed by the relative motions. Even with the added meandering that we implement (see Section 3.3, Fig. 1), the sampled plume resembles the shape of the source. Consequently, until the plume gets sufficiently mixed by smaller eddies, the variability between OTM33a

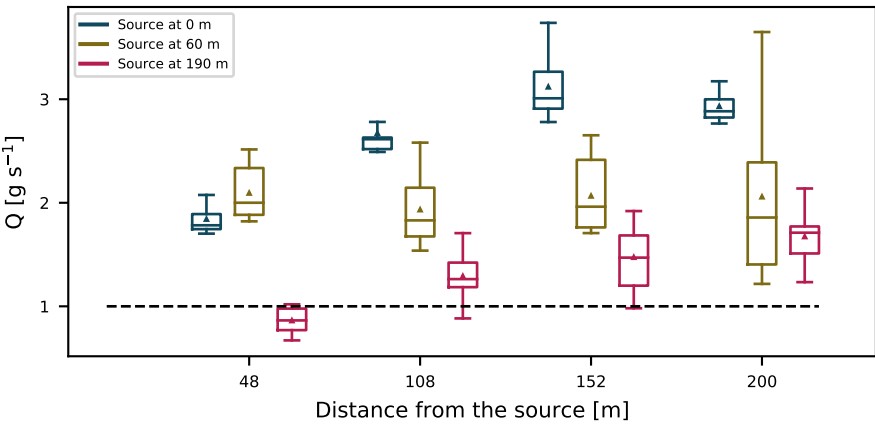

**Figure 8.** Estimates of the source strength using the OTM33a method for the emission heights 190 m, 60 m and 0 m at four different distances. Boxes show the interquartile range, while the whiskers span from 5 to 95 percentile of the data and show the mean and median. The dashed line refers to the true source strength used in the LES.

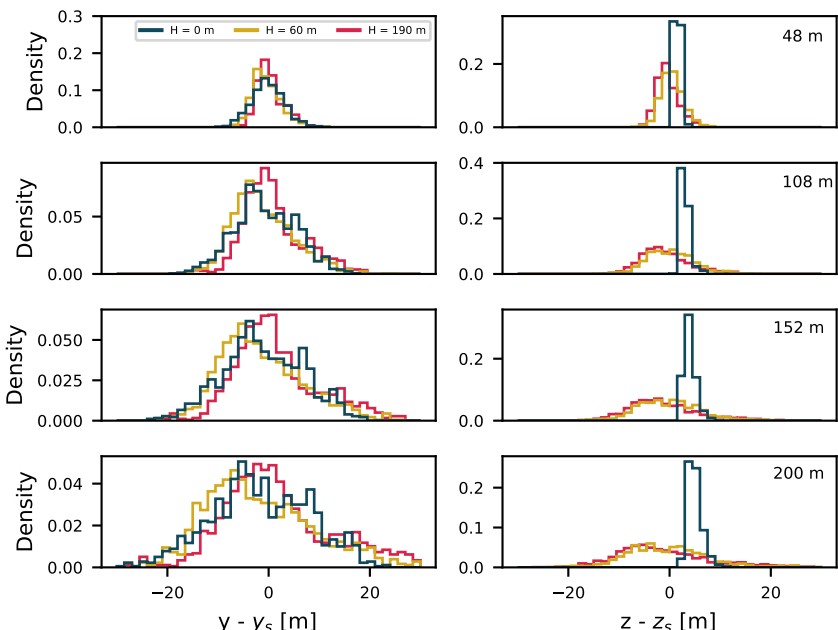

**Figure 9.** Probability density functions of the instantaneous plume centerline positions with respect to the plume emission positions in the y (left column) and z (right column) directions. Bins of 1.5 m were used.





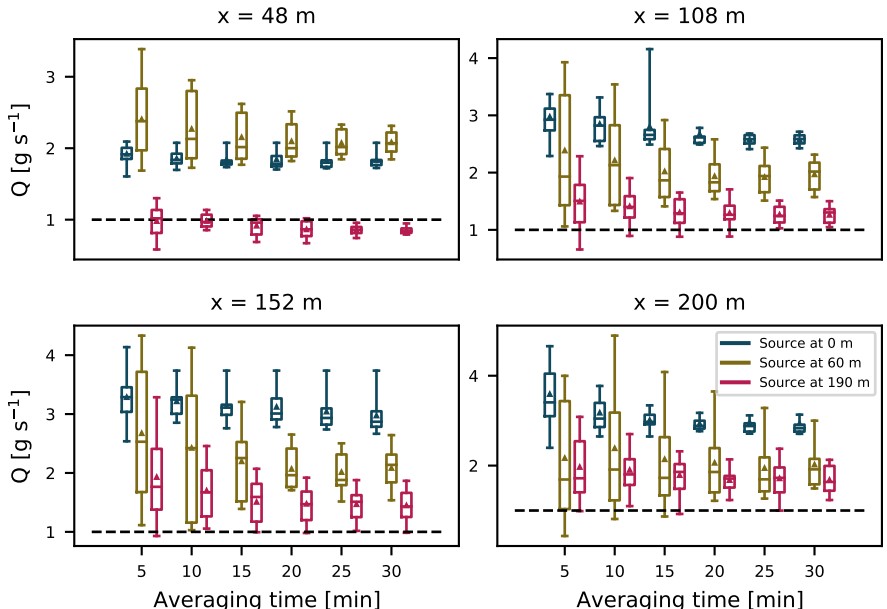

**Figure 10.** Source estimation using OTM33a method for 4 different distances from the source and for different averaging times. Boxes show the interquartile range, while the whiskers span from 5 to 95 percentile of the data and show the mean and median. The dashed lines refer to the true source strength used in the LES.

experiments used to produce the box-plots on Fig. 10 is not large. The increase in uncertainty with distance is related to the increase of the plume size.

### 4.3.1 Structure of the plume close to the source

To study the structure of the plumes close to the source, we analyze the plume statistics following the approach described in Section 3.4. To that end, we investigate the two processes responsible for plume dispersion (mixing and meandering) separately. The effect of turbulent mixing on the plume ($\sigma_{mix}$) is isolated by averaging the 3600 plumes after aligning them according to their (displaced) center of mass in y and z directions. Figure 11 shows these time averaged plumes, both aligned and non-aligned (meandering included) in the y direction at four distances from the source for the largest emission height. The figure also depicts Gaussian functions fitted to the aligned and non-aligned plumes, calculated according to Eqs. 9 ($\sigma_{mix}$) and 10, respectively. For reference, also the recommended OTM33A Gaussian dispersion is plotted.

In general, the non-aligned Gaussian functions are wider due to the meandering effect of the larger eddies that has been implemented (Eq. 9, $\sigma_{meander}$). As before, we find that the OTM33A dispersion coefficients are significantly larger than the time-averaged plumes. For the closest transect at 48 m, the tails of the non-aligned plume are very short and very similar to the aligned plume. This supports the observation that close to the source the plume is still very narrow and is not moved much by the larger sized eddies or dispersed around its centerline by the smaller ones. The shape of the plume at 48 m is determined by





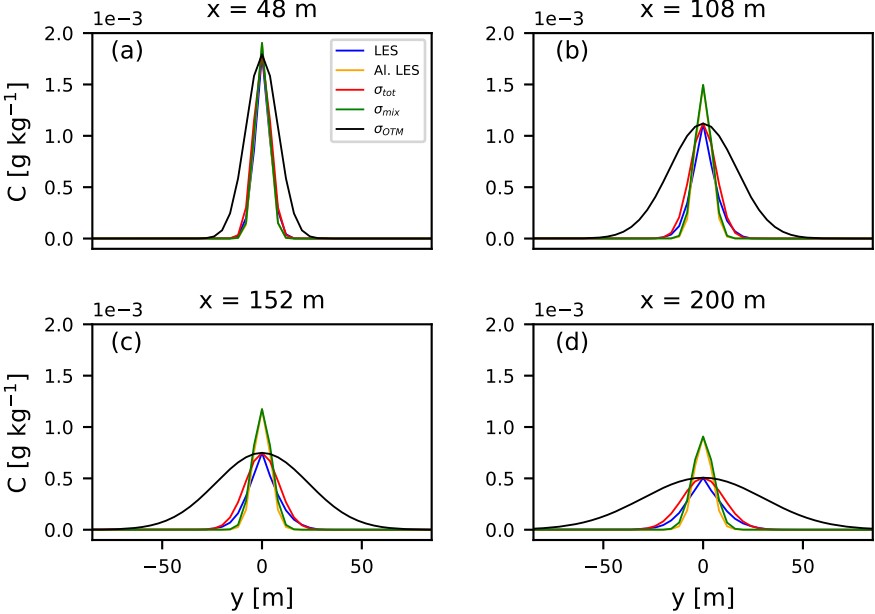

**Figure 11.** Time averages of instantaneous plumes (blue) on the plume centerline and the instantaneous plumes aligned according to their centers of mass (orange), in the y direction. Fitted through them are Gaussian functions with one standard deviation of their width $\sigma_i$ [m] (green: mixing; red: total). For reference, we show Gaussian functions fitted to the maximum of the non-aligned plume with $\sigma$ taken from OTM33A (black). Emission and sampling heights were 190 m.

370    the shape of the sources (section 3.1). Further downwind the difference between the aligned and non-aligned plumes grows, which indicates that the plume is being moved around significantly by larger eddies. At 200 m from the source, the aligned plume is still compact, which indicates that dispersion by small eddies is a slow process. The same behavior can be observed for the z direction (not shown). The values of the derived dispersion coefficients are given in Table 2.

We tested the OTM33a method with the dispersion coefficients derived from the LES. Understandably, the source estimates
375    improve largely compared to OTM33A method, but we still find estimation errors up to $\approx 40\,\%$. We argue that these errors are caused by the vertical displacement of the plume during transport (see Fig. 6). As previously mentioned, one of the assumptions of the OTM33a method is that the measurements are taken at the emission height. However, very close to the source the mean plume position, emission height, and the mode of the instantaneous plume positions do not necessarily coincide (Fig. 9). This is a likely consequence of the skewed vertical velocity field discussed previously in section 4.2. On top of that, we found that
380    the pdf of the instantaneous plume positions is also slightly skewed in the y direction (Fig. 9), which contributes to the overall error.

In conclusion, we find that the errors associated with the OTM33A method are sizeable. One source of errors is associated with the assumed dispersion coefficients, which were found to be too large compared to LES. Other sources of errors are related to assumptions made in the Gaussian plume model.





**Table 2.** Dispersion coefficients $\sigma$ (in m), obtained by using the definitions presented in section 3.4, for the plume emitted at 190 m.

| Distance [m] | $\sigma_{y,tot}$ | $\sigma_{y,mix}$ | $\sigma_{y,meand}$ | $\sigma_{z,tot}$ | $\sigma_{z,mix}$ | $\sigma_{z,meand}$ |
|---|---|---|---|---|---|---|
| 48 | 4.56 | 3.62 | 2.78 | 2.90 | 1.83 | 2.25 |
| 108 | 7.63 | 4.68 | 6.02 | 5.60 | 3.06 | 4.69 |
| 152 | 10.01 | 5.54 | 8.34 | 7.50 | 4.02 | 6.33 |
| 200 | 12.65 | 6.56 | 10.80 | 9.44 | 5.08 | 7.69 |

### 4.4 Source strength estimation from car measurements

Now we move to the analysis to more dispersed plumes that are sampled further away from the source (> 200 m). Figure 12 shows results of car measurements taken perpendicular to the mean wind and at plume emission height for all three sources. Estimates were made following the Inverse Gaussian method described in Section 2.3. The employed Gaussian plume model (Section 2.1) uses Briggs dispersion coefficients and the mean wind speed in the x direction at the height of the release. Estimates are shown for transects taken by an infinitely fast car. The transects are taken according to the sampling procedure described in Section (3.3.2). If only one transect is made, the estimated source strength shows a large spread for all distances, with estimates being up to 4 times larger than the real source strength. The medians show a negative bias indicating that a large fraction of the plumes has relatively low concentrations, and a few plumes exhibit (very) large concentrations. This result is expected, since the most concentrated part of the plume is moving over a 2D (y-z) plane (e.g. Fig 5). The probability of sampling this part of the plume with a 1-D transect is less likely. When the source strength is calculated using averaged plumes (see Section 3.3.2), it becomes more likely to estimate the emission strength. This is due to turbulent fluctuations in the plumes being averaged out. As a result, the averaged plume becomes more Gaussian as more transects are included in the average. For instance, the spread drops by $\approx 50\%$ if 10 transects are averaged. These results are in line with the findings of Caulton et al. (2018), who proposed averaging over at least 10 transects. As opposed to the estimations from the higher two plumes, the estimations from the lowest plume exhibit very little variability. As we have discussed in relation to Fig. 5, large eddies do reach the surface but merely displace the plume, and plume dispersion at this level is predominantly caused by small eddies processes.

In contrast to the OTM33A method, the estimated source strengths converge to the true value with sufficient averaging time using car transects. This is due to the fact that car transect method calculates the mass flux of the tracer by integrating over the (assumed) Gaussian profile. The flux of the mass through any given y-z plane in both models is conserved and equal, irrespective of the width of the actual plumes. Note here that the LES plume in our analysis is still much narrower in the y direction compared to the Gaussian plume. In the vertical direction, the LES plume width is comparable to the Gaussian (Fig. 7). If this would not be the case, the analysis would give incorrect source estimates, since a different displacement of mass in the vertical would lead to a different horizontal line integral.

It can be seen that the estimations, depending on their distance from source, converge to a slightly different value than the true source strength. This can be related to the position of the plume centerline and the plume mode discussed in sections 4.2

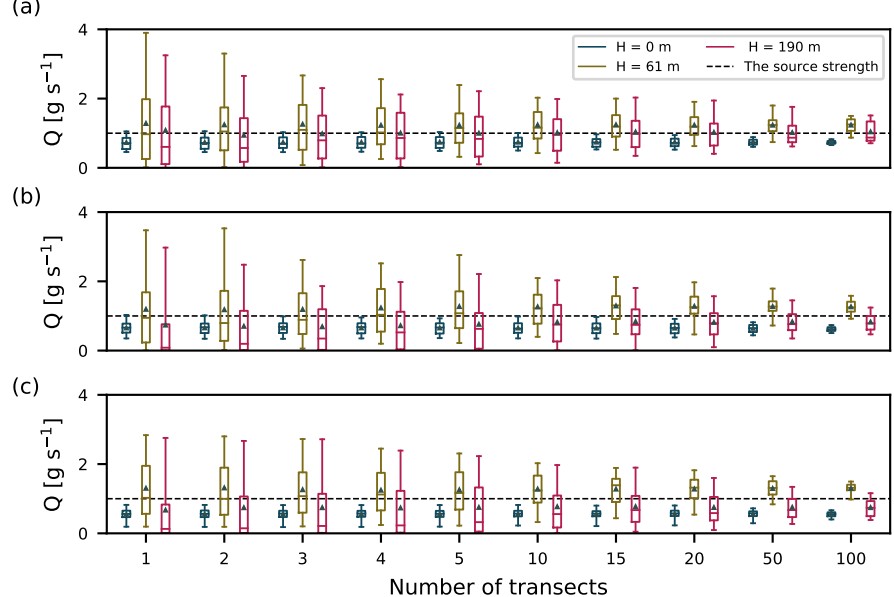

**Figure 12.** Estimates of the source strength from instantaneous plume transects for all three emission heights. Boxes show the interquartile range, while the whiskers span from 5 to 95 percentile of the data and show the mean and median. Distances from the source are (a) 200 m, (b) 624 m, (c) 1248 m.

and 4.3. For the LES plumes the position of the mode varies with the distance from the source while in the Gaussian plume model the plume centerline does not diverge from the emission height. This mismatch in the two models effectively means that two plumes with different emission heights are being compared. When the emission height in the Gaussian plume model is

adjusted to match the height of the LES plume mode for a certain downwind distance, the estimation error disappears.

Lastly, we have repeated the analysis of source strength estimations sampled outside of the plume centerline (not shown). Notably, the closer the plume is sampled to its edge, the more transects are needed for the estimates to converge to the true value. This was an expected result since at its edge the plume shows greatest variability as shown in previous studies (e.g. Dosio & de Arellano (2006); Gailis et al. (2007); Ardeshiri et al. (2020)).

To study the convergence of the results, Fig. 13 shows standard deviations of the source strength estimation for the plume emitted at 190 m. The standard deviations are shown for the first 40 plumes at 8 different distances from the source. After $\approx$ 10 transects, the standard deviation decays with the inverse square root of the number of averaged plumes. This confirms that all transects through the plume are independent, and that the time difference between the samples was not too short. We also studied the influence of driving speed through the plume for cars driving slow (4 m s$^{-1}$) and fast (12 m s$^{-1}$) but have found no

significant difference in results from transects taken by infinitely fast cars.





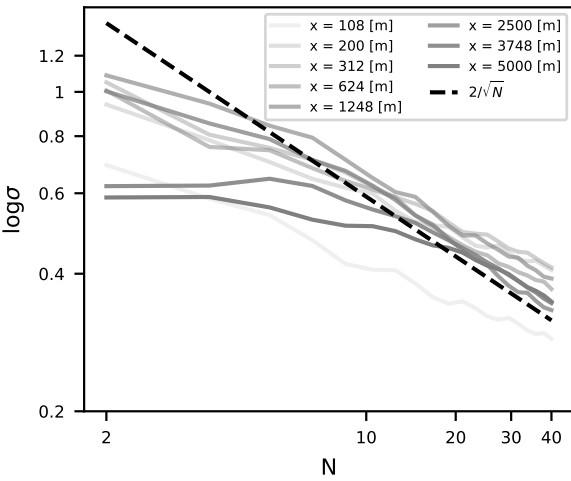

**Figure 13.** Standard deviation of the source strength estimates with increasing sample size for eight different distances from the source. The emission and sampling heights were 190 m.

## 5   Discussion and conclusion

In this study, we performed a large eddy simulation (LES) of point source plumes released into a neutrally stable, homogeneous and statistically stationary turbulent flow over a flat terrain. Simulations were performed following the laboratory experiment by Nironi et al. (2015) of point-source plume dispersion in a turbulent channel flow. Point sources were placed at three altitudes
$z = [0, 60, 190]$ m. We sampled our numerical plumes according to two measurement protocols that aim to estimate point source strengths with the aid of the Gaussian plume formalism. The aim was to quantify the uncertainties of the drive-through and OTM33a methods.

We found that the time-averaged LES plume has a Gaussian shape, but that the dispersion rate of the plume in the y direction is slower compared to a Gaussian plume with the Briggs dispersion coefficients representative of a neutral boundary layer
(e.g. Griffiths (1994)). One of the reasons why our y-dispersion is smaller might be the lack of large scale variability in our simulation, which is instead forced by a constant gradient pressure force. In the vertical, the discrepancies between the Briggs coefficients and the LES plumes were less pronounced. However, for smaller release heights, the mean plume centerline is displaced in the vertical, a feature that is not captured in the standard Gaussian plume dispersion coefficients. The rise of the plume centerline downwind from the source is caused by the nature of the boundary layer turbulence, which has large areas of
slow sinking motions, and small areas with stronger upward motions.

Application of the OTM33A method to our simulated plumes showed that we tend to overestimate the source strength by $\approx$ 50 - 200 %. Previous studies (Edie et al., 2020) showed a two-$\sigma$ uncertainty in the source strength of $\pm$ 70 %, but without a bias. The significant overestimation in our results is a direct consequence of the OTM33A formalism in which the derived source



strength depends linearly on the dispersion coefficients. Coefficients based on Pasquill-Gifford dispersion curves (U.S. EPA
(2014), e.g. Seineld (1986)) and Briggs coefficients (Griffiths, 1994) are both too dispersive compared to the LES simulation.

By aligning and averaging the plumes according to their center of mass we were able to show that, at distances smaller than
∼150 m from the source, the plume shows a shape similar to the source shape, i.e. a very narrow Gaussian. The aligned and non-
aligned plumes are similar indicating that the plume is moved very little from its center of mass by larger eddies (meandering)
even though OTM33A accounts for that. From Fig. 3 this is most obvious for the plume at 48 m. Further downwind the height of
the aligned plume peak gets smaller indicating the dispersion in the z direction also plays an important role. Nevertheless, if the
dispersion coefficients derived for the individual LES plumes are used in combination with the OTM33a method, significantly
smaller errors are found. Another source of the errors in the OTM33a method is the position of the plume in relation to its
centerline. The method assumes that the plume centerline position, emission height and the mode of the centerline position
coincide. We were able to show that this is not necessarily true and that this mismatch leads to additional uncertainties in the
source estimation.

We also simulated drive-by's at distances up to 1248 m from the source. The plumes were sampled simulating different car
speeds with a sampling frequency of 1 Hz to mimic realistic field conditions. We used the Inverse Gaussian Model (IGM)
method to derive the source strength, with the mean wind taken from the LES at release height and using Briggs dispersion
coefficients. We found that the correct source strength is estimated if the result is averaged over sufficient different realisations.
To estimate the source strength within ≈ 40%, we recommend to average over at least 15 drive-by's. This supports the findings
from Caulton et al. (2018) who recommended at least 10 transects. Our results show no significant influence of the driving
speed on the source strength estimation. The IGM method is insenitive to errors in y-dispersion, because the method depends
on the line integral in the y-direction. We found, however, that a mismatch between the vertical centerline position of the plume
and the emission height does produce an error in the source estimation. This error can be corrected by adjusting the height of
the Gaussian plume to match the simulated plume centerline.

Our study has shown some of the advantages and drawbacks of two commonly used measurement techniques for source
strength estimations. To arrive at our conclusions, we used the neutral channel flow experiment that resembles the purely
shear driven turbulence in an atmospheric surface layer. In this setting, the possible errors in the estimations are expected to
be minimized since the turbulence is well understood and the Gaussian plume model is logically derived. A next step would
be to repeat this study for different stability conditions in a idealized setting such as this or to re-create real field conditions
(Ražnjević et al., preprint). With constantly improving numerical techniques, LES is capable of reproducing real meteorological
conditions encountered in the field. Combined with the improving observational techniques, this approach is expected to lead
to better estimates of source strengths.

*Author contributions.* AR, CvH, and MK designed the research. AR performed the simulations and all analyses, and wrote the manuscript
in close collaboration with CvH and MK.





*Competing interests.* There are no competing interests.

*Acknowledgements.* This project is part of the Methane goes Mobile - Measurements and Modelling (MEMO$^2$) project. This project has received funding from the European Union's Horizon 2020 research and innovation programme under the Marie Sklodowska-Curie grant agreement No 722479. Maarten Krol received funding from the European Research Council (ERC) under the European Union's H2020
research and innovation programme under grant agreement No 742798. Authors acknowledge and thank Pietro Sallizoni and Massimo Marro for sharing their experimental data.

## Appendix A: Plume sampling procedure to mimic large-scale induced plume meandering

Here we describe the plume sampling procedure used to mimic the large-scale plume meandering necessary for the OTM33a method.

If we define 0 as the left edge of the plume and 1 as the right edge, we can define a function of time, $\zeta$, that oscillates between 0 and 1 with an uniform step, essentially mimicking forward and backward motions through the plume. In order to achieve denser sampling around the centerline, we re-define our sampling function in a way that gives us the relative position between -0.5 and 0.5, $\widehat{y}_i$, as:

$$\widehat{y}_i = \zeta_i + A(\tfrac{1}{2} - \zeta_i)(1 - \zeta_i)\zeta_i - \tfrac{1}{2}. \tag{A1}$$

Where $\widehat{y}_i$ is the grid point at which the sample was taken at the timestep $i$. Factor $A$ determines the density of the sampling points around the centerline, and we have set it to $A = 3$. We can then convert this array into dimensional units to find the position, $(\Delta y)_i$, at which the sample is taken, by adapting the relative position, $\widehat{y}_i$, to the actual plume width $L$ as is shown in Eq. A2

$$(\Delta y)_i = L\widehat{y}_i. \tag{A2}$$

The acquired sampling pattern for two distances from the source is shown in section 3.3.1, Fig. 1a. We applied the sampling strategy at x = [48, 108, 152, 200] m from the source. With the assumed $\theta = 15°$, the width over which plumes were sampled was $L = [21, 37, 49, 62]$ m for each of the distances from source respectively.



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
