# Peer review of "Evaluation of two common source estimation measurement strategies using large-eddy simulation of plume dispersion under neutral atmospheric conditions"

_Atmospheric Measurement Techniques, 2022_

## Referee Comment (RC1)

**Comments to "Evaluation of two common source estimation measurement strategies using large-eddy simulation of plume dispersion under neutral atmospheric conditions"**

1. Major Concerns

     a) In the introduction part, comparison between different existing techniques is not enough, need some improvement to highly the significance of your current study.

2. Minor Concerns

     1) line 35, "since they do not require direct access to the source". Argument is not strong since other popular techniques also has this feature.

     2)line 62, "turbulent channel flow". Is it more suitable to state here that we focus on atmospheric lower boundary layer that can be represented by channel flow. Otherwise, people may be confused by the ABL and channel flow.

     3)line 65," The LES represents perfect field conditions, without interference of confounding factors that may degrade the performance of source estimation methods under actual field conditions." The argument is weak, one of the reason you prefer virtual experiment via LES is it can serve as a benchmark that can be controlled ideally.

     4)line 71, "his". A typo, should be "this"?

     5)equation 1. Please indicates explicitly what is x, y, and z direction since people from different area may prefer different definition of y and z.  Also, the formula has no explicit dependence on x. Please change the variance either as a function of x, or in other explicit way.

     6)line 104. More precise, "averaged plume centerline" instead of just "plume centerline".

     7)line 189, "Onto the sampling angle, the instantaneous wind direction measured at each sampled point was added." What does this exactly mean?

     8)line 191, "delta y". Better use another symbol to avoid confusion with grid resolution.

     9)line 199, "The concentration measurements were taken perpendicular to the mean wind over the whole width of the domain". The data is reliable when they are close to boundary since you impose periodic boundary condition laterally. Artificial effects may be introduced for plume close to the boundary.

     10)line 205, "The same time delay was applied between each consecutive transect in an individual set". Not clear for how these procedures preformed, better with some illustrations.

     11) Figure 1 (c). dy symbol should be changed, refer to comments 8).

     12)line 220, "To separate these two processes". Not clear and how is it related to the definition you introduced following.

     13)line 245, "Despite the discrepancy in the mean wind, very good agreement is found between the wind speed variances (Fig. 2 b) and covariances (Fig. 2 c). Very good agreement is found for the triplet correlations as well (Fig. 2 d).". Any interpretation why mean value deviation is large, while the one for higher order statistics is small. This is a little bit contrary to typical cases we encountered in everyone's study, where the mean value deviation is small, but higher order statistics do not agree very well.

     14)line 250, "Following the good agreement of the higher order velocity statistics, we expect that the mixing of the plume in the crosswind directions is 250 well represented in the LES". This is indirect derivation, any direct quantity you can get to reach the same conclusion.

     15)line 255,"roughness velocity". Friction velocity?

     16)line 255, "delta". You define the turn over time based on boundary layer height, which is the effect of viscosity diffusion. Then why not the advection time scale?

     17) Figure 3. The agreement between your result and experiment one is because the position you select is close to the source, therefore the diffusion is dominant. What if you shift your virtual measurement point further downstream, where the advection effect is significant after the boundary layer of the scalar is fully developed. Also the scalar boundary layer height is different from fluid boundary layer height. Beside, please define m3 and m4 at somewhere.

     18)line 270, "To give the 270 reader an intuitive understanding of the spatial characteristics of the turbulence". Why turbulence, should that be the plume structure? Also, should that be the spatial distribution of the plume.

     19)Figure 4. Last subfigure has discontinuity for the plume concentration. Better to make it continuous by adjusting the color level range to avoid any confusion.

20)Figure 5. Any good reason that you put 100 instantaneous results here? People typically use mean + variance to get the same idea you want to show.

21)line 284, "an average". Please specify here whether it is spatial average or temporal average?

22)line 284, "over a large number of time steps". A large time window?

23)Figure 6. It is the isoline, right? Please specify it. The domain height is not the same for each case, better be the same for comparison.

24) line 296, "the width of the Gaussian plume and the highest LES plume only start to diverge far from the source". Why far field? Any reason?

25) line 302, "but are much smaller than the Briggs Gaussian plume coefficients". Why?

26) line 319, "Instead of emitting the tracer as a point source, we use a Gaussian distribution to pre-disperse the source". Suggest to conduct a comparative study to see how sensitive of the source strength estimation if you change the Gaussian kernel size, also, how about use HyperGaussian.

27) Figure 9. Please specify the line symbol in words in figure caption. Same for other figures if it is applicable.

28) Line 360. Reminder the reader where you get the deviation of mixing.

29) line 375, "estimation error up to". Please specify where is the corresponding figure.

30) line 380. Perhaps one reason is that your domain width is not large enough and the plume reach the boundary.

31) line 470, "different stability condition". One of the helpful references is the "Xiao, S., Peng, C., & Yang, D. (2021). Large-eddy simulation of bubble plume in stratified crossflow. Physical Review Fluids, 6(4), 044613." Where the plume structure is no longer to be a Gaussian shape vertically when the plume is developed underwater (corresponding to stable boundary layer in ABL).

---

## Referee Comment (RC2)

Review of the manuscript
"Evaluation of two common source estimation measurement strategies using large-eddy simulation of plume dispersion under neutral atmospheric conditions"

**General assessment**

This is an interesting study that uses the large-eddy simulation (LES) technique to test the validity of two common air quality measurements/models to estimate point source emission strength. Although the study finds substantial discrepancies between LES and measurements, it still highlights possible deficiencies in both approaches. I only have minor comments that I hope the authors would address.

1) I must note that the blame is mostly put on LES. On Line 245, the issue of the log-layer mismatch in LES is discussed as a possible reason for overshooting the mean velocity profile, but this is essentially undermining the use of LES. There are several studies that addressed this issue (besides from Brasseur who reported both undershoots and overshoots depending on a variety of parameters). See for instance Bou-Zeid et al. 2005; Physics of Fluids).

2) The authors argue in the introduction that DNS is becoming affordable and few paragraphs later mention that LES is expensive. It is fine to use LES in idealized conditions to test theoretical arguments, so I wouldn't undercut the approach.

3) The authors should at least comment/speculate on the effects of stability on their results. What do they expect in terms of statistics under unstable conditions?

4) The source characteristics need to be clarified. Emissions prescribed as a Gaussian distribution, as opposed to uniform source from one grid-cell mimicking a true point source, need to be discussed

---

## Author Comment (AC1)

**Authors responses to review 1**

Firstly, we would like to thank the reviewer for the time and effort they invested in helping us improve our text. In this document we will address their comments to the best of our abilities. Here the reviewers comments will be repeated in *italics* and our responses will be in plain text.

**Response to review 1**

**Major concerns**

*In the introduction part, comparison between different existing techniques is not enough, need some improvement to highly the significance of your current study.*
We changed the introduction in multiple places in order to make the point of significance of our study. We hope the text now has a better flow and there is a clear motivation for the study we did. The changes have been made as follows:
Line 35: "For the detection and quantification of local sources, techniques using mobile platforms are particularly useful, since they allow for large areas being covered with measurements in relatively short time period and quick source strength estimations using simple models."
A new paragraph has been added after line 52: "Both measurement techniques, as previously mentioned, rely on the Gaussian plume model to estimate emission rates. The Gaussian model is the solution to the advection-dispersion equation for a point source with the assumption of constant wind and dispersion coefficients that are functions of downwind distance and atmospheric stability (e.g. Seinfeld (1980)). As such, the methods compare the modeled stationary plume with the measured turbulent plume. Such comparison is bound to lead to estimation errors, unless enough measurements have been collected to average out the atmospheric variability. Therefore, a systematic and controlled study is needed to constrain the influence of turbulence on these measurement techniques. Apart from Caulton (2018) who analysed the car transect method using LES and concluded that at least 10 transects are needed to average out the variability, such study has not been conducted to the best of our knowledge. "

**Minor concerns**

*1) line 35, "since they do not require direct access to the source". Argument is not strong since other popular techniques also has this feature.*
The reviewer is correct here. The argument does also apply to other source estimation techniques. We have changed the argument to: "For the detection and quantification of local sources, techniques using mobile platforms are particularly useful, since they allow for large areas being covered with measurements in relatively short time period and quick source strength estimations using simple models."

*2) line 62, "turbulent channel flow". Is it more suitable to state here that we focus on atmospheric lower boundary layer that can be represented by channel flow. Otherwise, people may be confused by the ABL and channel flow.*
Agreed. The lines 61 - 62 have now been re-written to say: "Due to the high computational costs involved in LES, we limit this study to the lower atmospheric boundary layer under neutral conditions. The neutral atmospheric surface layer can be well represented by a turbulent channel flow and is one of the most canonical and well-studied cases of atmospheric turbulence."

*3) line 65,"The LES represents perfect field conditions, without interference of confounding factors that may degrade the performance of source estimation methods under actual field conditions." The argument is weak, one of the reason you prefer virtual experiment via LES is it can serve as a benchmark that can be controlled ideally.*
The LES being an ideal experiment that was completely controllable was the message of the sentence. We have tried to formulate it more clearly: "The LES represents the ideal experiment in which all sources of uncertainties are controllable and quantifiable. By using LES we are able to study the influence of turbulent fluctuations on plume dispersion and consequently on the measured plume, which can be used as a benchmark for future measurement campaigns."

*4) line 71, "his". A typo, should be "this"?*
Yes. Fixed.

*5) equation 1. Please indicates explicitly what is x, y, and z direction since people from different area may prefer different definition of y and z. Also, the formula has no explicit dependence on x. Please change the variance either as a function of x, or in other explicit way.*
The explicit x dependence has been added to the equation (1) such that the variances are now written as functions of x. The definition of direction was added in the text to line 84: "Here, x direction is defined as the direction of the mean wind $\bar{u}$, y is the horizontal crosswind direction and z points away from the surface."

*6) line 104. More precise, "averaged plume centerline" instead of just "plume centerline".*
Corrected.

*7) line 189, "Onto the sampling angle, the instantaneous wind direction measured at each sampled point was added." What does this exactly mean?*
The meandering was added to the simulation by moving the sampling point through the plume on the $\theta = [-15°, 15°]$ angle. Here $\theta = 0°$ corresponds with y position directly downwind from the source. The $\theta$ angle in this cases represents a plume meandering that would happen due to large scale forcing that we have taken to be constant and the amplitude of these oscillations did not change for the duration of the experiment. However, the instantaneous wind angle is also subjected to turbulent wind fluctuations. In order to have them still represented in the method, we calculated the angle of the instantaneous wind for each step through the plume ($Y_i$ in the equation (A2) in Appendix A) and added that number to the angle $\theta$. We think the explanation in the text sufficiently explains this.

*8) line 191, "delta y". Better use another symbol to avoid confusion with grid resolution.*

The symbol $Y$ is now used instead of $(\Delta y)_i$ in line 191. The correction has also been done in the Appendix A equation (A2) and line 492.

*9) line 199, "The concentration measurements were taken perpendicular to the mean wind over the whole width of the domain". The data is reliable when they are close to boundary since you impose periodic boundary condition laterally. Artificial effects may be introduced for plume close to the boundary.*

This is true. However, in our simulation a boundary condition that prevents the plume from re-entering the domain through the boundaries has been added on all lateral boundaries. Apart from that, our plume is not fluctuating on a wide enough angle in the lateral direction to reach the boundaries even far downwind. This is why we had to impose it artificially for the use of OTM33A method.

*10) line 205, "The same time delay was applied between each consecutive transect in an individual set". Not clear for how these procedures preformed, better with some illustrations.*

The language we used is a bit unclear it would seem. The meaning of line 205 is that the transects are taken at regular time intervals and that same time delay was applied before taking a new set of transects. The lines 204 and 205 have been changed to: "The time delay between each set of measurements was taken as 10 s. The transects have also been sampled with a 10 s delay between them."

*11) Figure 1 (c). dy symbol should be changed, refer to comments 8).*

Changed. The symbol now used is $Y$.

*12) line 220, "To separate these two processes". Not clear and how is it related to the definition you introduced following.*

We have re-written the line 220 as: "To separate the influence of plume meandering from the influence of relative dispersion on the total plume growth, we can define relevant plume metrics in absolute co-ordinate system (i.e. in relation to the ground) and relative coordinate system (i.e. dispersion around instantaneous center of mass). First, the center of mass of the instantaneous plume in the relation to the ground, $z_m$, on its y-z transect is defined as:"

*13) line 245, "Despite the discrepancy in the mean wind, very good agreement is found between the wind speed variances (Fig. 2 b) and covariances (Fig. 2 c). Very good agreement is found for the triplet correlations as well(Fig. 2 d).". Any interpretation why mean value deviation is large, while the one for higher order statistics is small. This is a little bit contrary to typical cases we encountered in everyone's study, where the mean value deviation is small, but higher order statistics do not agree very well.*

We believe our wording was the main problem here again. We do not mean that the LES produced a bad mean velocity profile, the profile fits the observation still quite well. The variations have been reproduced by LES so well that by comparison the difference in the mean wind seems relatively large. Similar result as ours can be seen in Ardeshiri et al. (2020). On their Fig. 1 a) the mismatch between the observations and the LES can be seen as well as a very nice fit in the variance profiles. The text was altered as: "Despite the slight discrepancy in the mean wind, very good agreement is found between the wind speed variances (Fig. 2 b) and covariances (Fig. 2 c). Very good agreement is found for the triplet correlations as well (Fig. 2 d). Ardeshiri et al. (2020) also reproduced the Nironi et al. (2015) case using LES, and have shown very similar results to the ones shown here."

*14) line 250, "Following the good agreement of the higher order velocity statistics, we expect that the mixing of the plume in the crosswind directions is 250 well represented in the LES". This is indirect derivation, any direct quantity you can get to reach the same conclusion.*

It is indirect in this sentence yes. However, in the same section we show the plume statistics compared between the experiment and LES. There it is visible that the widths of both plumes are very similar confirming our statement from above.

*15) line 255,"roughness velocity". Friction velocity?*

Friction velocity indeed. Fixed.

*16) line 255, "delta". You define the turn over time based on boundary layer height, which is the effect of viscosity diffusion. Then why not the advection time scale?*

We use the boundary layer height since we are interested in the spatial evolution of the plume and not purely in the plume downwind displacement. The time it takes a typical eddy to travel to the top of the boundary layer and back does not necessarily coincide with the time it takes to travel over the advection lengthscale. That is why we opted for the boundary layer height in our definition of overturn time.

*17) Figure 3. The agreement between your result and experiment one is because the position you select is close to the source, therefore the diffusion is dominant. What if you shift your virtual measurement point further downstream, where the advection effect is significant after the boundary layer of the scalar is fully developed. Also the scalar boundary layer height is different from fluid boundary layer height. Beside, please define m3 and m4 at somewhere.*

We have shown results for the distance of 2.5 boundary layer heights ($\delta$) away from the source. That distance is about in the middle from all the available transects from the measurements where the closest transects was taken at $0.3125\delta$ and the furthest at $5\delta$. At that distance we expect neither the diffusion nor advection to be dominant. Nevertheless, despite the different processes that drive the dispersion on different distances from the source, we found that LES performed well on all distances. We attach here a figure that shows centerline values of all four moments at the emission height of each of the plumes and at several downwind distances.

We have defined $m_3$ and $m_4$ in 267: "The same can be observed for the other two moments shown here, the skewness $m_3^*$ and kurtosis $m_4^*$ (Fig. 3 bottom two rows)."

[Figure]

*18) line 270, "To give the 270 reader an intuitive understanding of the spatial characteristics of the turbulence".*

*Why turbulence, should that be the plume structure? Also, should that be the spatial distribution of the plume.*
We meant that turbulence is dictating the spatial distribution of the plume in the sense that the plume does not have the gaussian shape. But yes, we agree with the reviewers suggestion being more intuitive and the sentence now reads: "To give the reader an intuitive understanding of the spatial distribution of the plumes, Figure 4 shows instantaneous x-z cross-sections of the three simulated plumes at the y position of the source ($y_s$)."

*19) Figure 4. Last subfigure has discontinuity for the plume concentration. Better to make it continuous by adjusting the color level range to avoid any confusion.*
The colorscale on all three subfigures have been set to the min and max of the respective plumes. The discontinuity in the last subfigure comes from the patchiness of the plume at that snapshot. The figure shows the plumes on the x-z transect taken at the y position of the sources (y = 0), the plume on the last subfigure has been moved from y = 0 to the sides by turbulent motions. We have purposefully chosen this time instant to illustrate different behaviors plumes exhibit.

*20) Figure 5. Any good reason that you put 100 instantaneous results here? People typically use mean + variance to get the same idea you want to show.*
This is another illustrative figure. We could have used the mean and the variance, however they miss the information about the outliers. For the purpose of this study, where the methods are focused on evaluating emissions based on one dimensional transects, we found useful to explicitly show plume variability on a single transect.

*21) line 284, "an average". Please specify here whether it is spatial average or temporal average?*
It is temporal average. Fixed.

*22) line 284, "over a large number of time steps". A large time window?*
There is some confusion on our part about this comment. A large time window yes, but assuming nothing changes in the experiment meteorology wise, the averaging can be done over a large amount of samples taken at random time steps. The averaging does not have to be done on consecutive samples.

*23) Figure 6. It is the isoline, right? Please specify it. The domain height is not the same for each case, better be the same for comparison.*
Yes. The caption on Fig. 6 now reads: "Time averaged x-z cross-section of the LES plume at (top) 0 m, (middle) 60 m and (bottom) 190 m height. The concentration fields are averaged over half an hour. The isolines are connecting the areas with the same concentration shown here in [g kg$^{-1}$]. Also plotted are the plume edges of the Gaussian plume (assuming Briggs diffusion coefficients) and the LES plumes. Centerlines $\overline{z_m}$ are plotted as dashed lines."
In the revised figure the domain heights are now all the same.

*24) line 296, "the width of the Gaussian plume and the highest LES plume only start to diverge far from the source". Why far field? Any reason?*
A possible reason for divergence of dispersion coefficients calculated using LES plumes from the Briggs coefficients could be in the fact that the divergence starts happening only when the mean plume reaches the ground. From the other two plumes it is also visible that influence of the ground on the mean plume

is not well represented in the Gaussian plume model.

*25) line 302, "but are much smaller than the Briggs Gaussian plume coefficients". Why?*
The Briggs coefficients have been calculated based on a release experiment. That means that the full range of motions that govern the plume dispersion were captured in their measurements on that day. In our case the largest motions that could have developed in our simulation were restricted by the size of the domain. This is the reason for artificial meandering we imposed through plume sampling discussed in Section 3.3.1.

*26) line 319, "Instead of emitting the tracer as a point source, we use a Gaussian distribution to pre-disperse the source". Suggest to conduct a comparative study to see how sensitive of the source strength estimation if you change the Gaussian kernel size, also, how about use HyperGaussian.*
The following has been added to the line 321: "Following this, we expect the different source size to have a different effect on the source strength estimation on distances very close to source. In future studies, it is recommended to quantify the effect of the prescribed source size on the source estimation using OTM33a." We used the Gaussian shape for the simplicity of implementation to the MicroHH code.

*27) Figure 9. Please specify the line symbol in words in figure caption. Same for other figures if it is applicable.*
The caption has been changed to: "Probability density functions of the instantaneous plume centerline positions with respect to the plume emission positions (blue: 0 m, yellow: 60 m, pink: 190 m) in the y (left column) and z (right column) directions. Bins of 1.5 m were used."

*28) Line 360. Reminder the reader where you get the deviation of mixing.*
We have some trouble understanding this comment. But we assume the reviewer meant *definition* instead of *deviation*. With this assumption, we changed the line 360 to: "The effect of turbulent mixing on the plume ($\sigma_{mix}$, as defined in section 3.4, eq. 9) is isolated by averaging the 3600 plumes after aligning them according to their (displaced) center of mass in y and z directions."

*29) line 375, "estimation error up to". Please specify where is the corresponding figure.*
We have not put the corresponding figure into the paper since it only shows that the OTM33A method works better (but not perfect) when the optimized dispersion coefficients are used. We are aware that the dispersion coefficients cannot be known for every plume measured and we do not claim using the coefficients from our study would necessarily improve estimates from the field measurements. Our main message here was that the method could work under very specific conditions, and that it is very sensitive to the choice of the dispersion coefficients so in order not to confuse the reader we left the figure out but we added "(not shown)" to the text for clarity.

*30) line 380. Perhaps one reason is that your domain width is not large enough and the plume reach the boundary.*
There is a boundary condition on all lateral boundaries that prevents the plume from re-entering on the other side but permits the velocity components to pass through. So we do not believe that is the case. Apart from this, the furthest downwind where we looked at the pdf of centerline positions was 200 m, at this distance the plume is less than 100 m wide (visible on Figure 11.) and our domain width is 1500 m.

*31) line 470, "different stability condition". One of the helpful references is the "Xiao, S., Peng, C., Yang, D.*

*(2021). Large-eddy simulation of bubble plume in stratified crossflow. Physical Review Fluids, 6(4), 044613."*
*Where the plume structure is no longer to be a Gaussian shape vertically when the plume is developed underwater (corresponding to stable boundary layer in ABL).*

Yes, it would be nice to re-create the same experiment with some validations for the stable layer. Reference added.

---

## Author Comment (AC2)

**Authors responses to review 2**

We would first like to than the reviewer for their kind words and time invested in our study. We will do our best to address all of their comments in the text below. In this document, we have repeated the reviewers comments in the *italics* and our responses are in the standard font.

*This is an interesting study that uses the large-eddy simulation (LES) technique to test the validity of two common air quality measurements/models to estimate point source emission strength. Although the study finds substantial discrepancies between LES and measurements, it still highlights possible deficiencies in both approaches. I only have minor comments that I hope the authors would address.*

*1) I must note that the blame is mostly put on LES. On Line 245, the issue of the log-layer mismatching LES is discussed as a possible reason for overshooting the mean velocity profile, but this is essentially undermining the use of LES. There are several studies that addressed this issue (besides from Brasseur who reported both undershoots and overshoots depending on a variety of parameters). See for instance Bou-Zeid et al. 2005; Physics of Fluids).*
We indeed did take the measured profile as the truth and blamed the deviation from the log-layer on LES. We attach here a figure that shows velocity and variance profiles (matching the first two panels in Fig. 2) for Nironis channel experiment, an LES with the same horizontal resolution as the one in this paper, and a DNS with moderate Reynolds number, Re$_\tau$ = 1828. On this figure the LES and DNS match quite well and LES has slightly different mean wind profile than the one presented in the paper. The only difference between the LES from our paper and on this figure is the resolution in the z direction where the simulation in the paper has more grid points in the vertical (360) than the one in the figure (240). Following this, the blame for the overshoot in the mean velocity profile does seem to be with LES. However, after accounting for the difference in the mean winds between the LES and the experiment, the statistical profiles of the plumes (Fig. 3 ) match quite nicely. So yes, the LES has an overshoot in the mean velocity, but for our study that is not important since we use the experiment only for the validation of plume statistics and not for the analysis of the measurement techniques.

[Figure]

*2) The authors argue in the introduction that DNS is becoming affordable and few paragraphs later mention that LES is expensive. It is fine to use LES in idealized conditions to test theoretical arguments, so I wouldn't under cut the approach.*

Yes, we see how the way we formulated those sentences might raise a question of why are we not using DNS then. Now line 57 to the end of the paragraph reads: "DNS, as it resolves all details of the flow, would be ideal approach for studying plume dispersion, however, due to unfeasible computing costs involved, it cannot reproduce high Reynolds number flows (Pope, 2000). Nevertheless, in recent years it is becoming more affordable for atmospheric studies (e.g. Branford et al. (2011), Oskouie et al. (2017)), as computers have sufficient power to simulate atmospheric boundary layers with statistics that are slowly becoming Reynolds number independent."

*3) The authors should at least comment/speculate on the effects of stability on their results. What do they expect in terms of statistics under unstable conditions?*

Agreed. We have added the following paragraph to the Discussion section: "The plumes studied here were emitted into the neutral channel flow as it is the most canonical case of the atmospheric turbulence. Similar study should be performed for unstable and stable conditions. Based on our findings, we expect the unstable conditions to add to the variance of the plume as bouyancy effect adds to the production of turbulence. Conversely, for a stable atmosphere, we expect shorter averaging time (less plume transects) would be required to achieve 40 % accuracy."

*4) The source characteristics need to be clarified. Emissions prescribed as a Gaussian distribution, as opposed to uniform source from one grid-cell mimicking a true point source, need to be discussed.*

We did comment on the need of using a 3D Gaussian as opposed to emitting from a single grid cell in Section 3.1 where we discuss the implementation of sources. But we have opted for a dispersed source in order to avoid sharp gradient of scalar concentrations that would appear at the position of the source and could lead to numerical instabilities. Furthermore, Ardeshiri et al. (2020) in their LES study found that the sources have to be resolved over at least $4^3$ grid nodes to have scalar variance statistics converge.